# Coupling Effect of Expansion Agent and Internal Curing Aggregate on Shrinkage of High-Modulus Ultra-High-Performance Concrete

**Min Zhou** [1], **Tengyu Yang** [1], **Jinhui Li** [2,*], **Bing Qiu** [1], **Wenjun Qiu** [1], **Dongdong Chen** [3], **Baiyun Li** [1], **Benan Shu** [1], **Changsheng Zhou** [3], **Lixian Guo** [1], **Zi Yu** [2] and **Yongling Li** [1]

[1] Foshan Transportation Science and Technology Co., Ltd., Foshan 528000, China; shuba0411@126.com (M.Z.); yangtengyu@126.com (T.Y.); 13902598673@139.com (B.Q.); 17620132531@163.com (W.Q.); libaiyun1996@163.com (B.L.); shuba@whut.edu.cn (B.S.); jumo731374828@163.com (L.G.); lyling19941209@163.com (Y.L.)

[2] College of Materials Science and Engineering, Wuhan Textile University, Wuhan 430200, China; yu1018541283@163.com

[3] School of Materials and Science Engineering, Wuhan University of Technology, Wuhan 430070, China; chendongdong0714@163.com (D.C.); 18982816083@163.com (C.Z.)

[*] Correspondence: jhli@wtu.edu.cn

**Abstract:** In the realm of bridge structural engineering, it is customary to meticulously contemplate the material's strength and rigidity attributes during the dimensioning phase. In recent years, there has been a burgeoning interest in employing Ultrahigh-Performance Concrete (abbreviated as UHPC) for the construction of bridge decks and wet joints. However, the large self-shrinkage of UHPC can easily lead to shrinkage cracking and affect its service life. This study delves into the utilization of a blend of basalt coarse aggregate and high-modulus aggregate (HMA) in the formulation of Ultrahigh-Performance Concrete (UHPC) with the objectives of achieving exceptional strength (>180 MPa), superior modulus of elasticity (>56 GPa), and synergistic effect of using prewetted internal curing aggregate (ICA), metallurgical ore sand (MOS), and calcium–magnesium composite-based expansion agent (EA) to reduce the shrinkage of UHPC. Furthermore, the mechanical properties, shrinkage, hydration process, and microstructure of UHPC prepared with EA and ICA were studied. The results show that UHPC prepared with both 3% EA and 20% ICA had the optimal volume stability (the shrinkage was only 273 με at 180 d). In contrast, the 180 d shrinkage of UHPC with 3% EA and 20% ICA separately was 287 με and 373 με, respectively. In addition, the incorporation of EA and ICA can effectively improve the flexural strength of UHPC, although it affects the compressive strength and modulus of elasticity of UHPC (small decrease).

**Keywords:** UHPC; shrinkage; mechanical properties; expansion agent; internal curing aggregate; microstructure





## 1. Introduction

Ultrahigh-Performance Concrete (UHPC), as an emerging cement-based composite construction material, has excellent mechanical properties and durability that make it widely used in road and bridge projects [1]. The high modulus of elasticity guarantees that UHPC bridge deck panels, even with reduced thickness, will not exhibit substantial deformation when subjected to external forces, thereby optimizing the outstanding mechanical characteristics inherent to UHPC [2]. Due to the extremely low water–cement ratio and high cementitious material admixture of UHPC, its early self-shrinkage is much larger than that of normal concrete [3]. Self-shrinkage is considered to be a macroscopic volume change induced by capillary pressure due to the chemical processes of self-drying and cement hydration [4]. According to reports, self-shrinkage manifests in concrete when the water–cement ratio is approximately less than 0.42 [5] and escalates as the water-to-cement

ratio decreases. The early occurrence of high self-shrinkage can lead to potential cracking, which can affect the mechanical properties and durability of the hardened cementitious material. In order to decrease the amount of cementitious material, the high-modulus UHPC reduces its shrinkage by mixing with coarse aggregates However, in order to ensure the quality and lifetime of the project, it is still necessary to further reduce the shrinkage of UHPC and improve its volumetric stability.

At present, several approaches have been explored to reduce the shrinkage of UHPC, including the use of shrinkage reducers, incorporation of expansion agents, and mixing of internal maintenance aggregates [6–8]. SRA (Shrinkage-Reducing Admixture) could reduce the possibility of shrinkage-induced stresses, thereby mitigating the risk of cracking. SRA comprises nonionic organic surfactants, which effectively lower the surface tension of the pore solution. Consequently, this reduction in surface tension serves to reduce capillary stress during the process of water loss [9]. The decrease in capillary stress can reduce early drying shrinkage and both plastic and autogenous shrinkage [10]. The addition of SRA also reduces the water evaporation thus reducing the residual stresses [10]. Due to the SRA, the water evaporation is reduced and can maintain a high internal relative humidity [11]. However, several side effects associated with SRA usage have been reported, including reduced cement hydration rate, loss of entrained air, delayed setting time, and impact on early mechanical property development [12].

The chemical reaction between the expansion agent (EA) and water results in macroscopic volume expansion. Consequently, the early hydration of concrete produces a large amount of calcium alumina or calcium hydroxide. Furthermore, the expansion force produced by the crystallization and expansion pressure compensates for part of the self-shrinkage of the cementitious material [13]. Based on the primary components utilized, they are categorized into CaO-based, calcium aluminate-based, and MgO-based types [12]. The use of CaO-based EA can lead to rapid hydration of CaO [14]. Moreover, the addition of CaO-based EA also enhances the interfacial zone, especially in the early stages of hydration. Meanwhile, the EA also improves the concrete pore structure, thereby effectively mitigating drying shrinkage [15]. In contrast to traditional expansion agents, the expansion properties of MgO-based expansion agents are contingent upon the calcination temperature and residence time [16]. Residence time denotes the duration of calcination for the finely ground magnesite in the electric furnace. The residence time is the calcination time of the ground magnesite in the electric furnace. MgO-based expansion agents do not react with expansion until the concrete is hardened, and expansion has a low water requirement [17]. However, the use of EA can hinder the hydration of cementitious materials, especially in low water–cement ratio systems, which would reduce the free water available for cement hydration [18]. Su et al. [19] noted that the addition of a 2% calcium–magnesium compound EA to UHPC at a water-to-cement ratio of 0.2 resulted in a 10% decrease in its 28 d (concrete at 28 days denotes the condition of concrete following a span of 28 days of curing and consolidation) compressive strength.

Using internal curing aggregates (ICAs) to create supplemental moisture maintenance inside the UHPC also reduces shrinkage. Good candidates for internal curing aggregates should have high water absorption capacity. Moreover, it can easily release the absorbed water into the cementitious base before the relative humidity is about to drop. Meng [20] reported that increasing the volumetric inclusion of prewetted lightweight aggregates from 0% to 25% resulted in a 35% reduction in the self-shrinkage of UHPC with a water-to-cement ratio of 0.2. In addition, increasing the amount of internal curing aggregate can continue to reduce the self-shrinkage of UHPC. However, due to the low mechanical properties of the porous prewetted light aggregate, the compressive strength of UHPC is reduced by 25% at a prewetted light aggregate volume admixture of 75%. Some researchers found that the use of appropriate content of porous aggregates not only reduced self-shrinkage but also slightly increased the compressive strength of UHPC [21,22]. This can be explained by the formation of a thin and dense interfacial transition zone (ITZ) between the cement paste matrix and the porous aggregates.

The joint effect of EA and ICA reduces the self-shrinkage of UHPC when contrasted with the implementation of a single shrinkage reduction strategy (this may be a more effective solution). The use of ICA in addition to providing internal curing also allowed the EA to hydrate and expand more fully [23]. Valipour [24] claimed that UHPC prepared with 7.5% intumescent and 60% prewetted lightweight aggregate had a shrinkage of only 273 με at 28 d, whereas UHPC with 7.5% intumescent and 60% prewetted lightweight aggregate alone had a shrinkage of 350 με and 580 με at 28 d, respectively. Most of the previous studies used prewetted light aggregates as the ICA to reduce the shrinkage of UHPC, but their low aggregate strength will reduce the mechanical properties of UHPC. In this study, metallurgical ore sand (MOS) with multiple microfine-connected pores and high strength was selected. After being prewetted with full water, it replaced part of the high-modulus aggregate (HMA) in equal volume as the ICA. The high absorption rate and stable water storage of MOS were utilized to alleviate the shrinkage of UHPC by continuously releasing water after molding and hardening. Furthermore, the mechanism of the synergistic effect of EA and ICA on the mechanical properties and volume stability of high-modulus UHPC was studied for providing a theoretical basis for the promotion and application of UHPC in bridge engineering.

## 2. Experimental Procedure

### 2.1. Materials

P·II 52.5 silicate cement (CEM) was used for cement, with an apparent density of 3.15 g/cm$^3$. The specific surface area of silica fume (SF) was 18,300 m$^2$/kg, and the SiO$_2$ content was 90%. The fly ash microsphere (FAM) 28 d activity index was 113%. The expansion agent was CaO–MgO compound. HMA and MOS were selected as fine aggregates. HMA had an apparent density of 2530 kg/m$^3$ and a saturated surface dry water absorption rate of 6.8%, whereas MOS had an apparent density of 3100 kg/m$^3$ and a saturated surface dry water absorption rate of 10%. The chemical compositions of HMA and MOS are listed in Table 1. The coarse aggregate (C.A) was basalt crushed stone with a particle size of 5–8 mm, crushing value of 12%, and apparent density of 2900 kg/m$^3$, and the particle size distribution curve is shown in Figure 1. The superplasticizer is polycarboxylic acid high-performance superplasticizer agent with 40% solid content and 35% water reduction rate. Short straight steel fibers were used to enhance bending toughness with an equivalent diameter of 0.2 mm, a nominal length of 13 mm, and a density of 7850 kg/m$^3$.

**Table 1.** Chemical compositions of fine aggregates (wt%).

| Fine Aggregate | SiO$_2$ | Al$_2$O$_3$ | Fe$_2$O$_3$ | CaO | MgO | V$_2$O$_5$ | K$_2$O | FeO | TiO$_2$ |
|---|---|---|---|---|---|---|---|---|---|
| HMA | 39.1 | 52.3 | 1.8 | 1.05 | 0.27 | - | 0.61 | - | 2.46 |
| MOS | 18.1 | 14.9 | 5.2 | 24.6 | 8.4 | 0.22 | - | 2.55 | 22.3 |

### 2.2. Test Programs

According to previous studies, 8 mixing ratios were used to investigate the effect of EA and ICA on the performance of UHPC (shown in Table 2). Among them, group N is the reference control group, group E3 illustrates 3% of the EA in equal mass proportion to replace the cementitious material, group M10 illustrates 10% of the prewetted MOS in equal volume proportion to replace the HMA, and group E3M20 is mixed with both 3% of the expansion agent and 20% of MOS.

**Table 2.** Mixture proportions (kg/m$^3$).

| Mix ID | Cement | SF | FAM | EA | HMA | MOS | C.A | Fiber | SP | Water |
|---|---|---|---|---|---|---|---|---|---|---|
| N | 625 | 165 | 135 | 0 | 800 | 0 | 500 | 160 | 22.2 | 166 |
| E3 | 606 | 160 | 132 | 28 | 800 | 0 | 500 | 160 | 22.2 | 166 |
| E5 | 594 | 157 | 128 | 46 | 800 | 0 | 500 | 160 | 22.2 | 166 |

**Table 2.** *Cont.*

| Mix ID | Cement | SF | FAM | EA | HMA | MOS | C.A | Fiber | SP | Water |
|--------|--------|-----|-----|----|-----|-----|-----|-------|------|-------|
| E8 | 575 | 152 | 124 | 74 | 800 | 0 | 500 | 160 | 22.2 | 166 |
| M10 | 625 | 165 | 135 | 0 | 720 | 98 | 500 | 160 | 22.2 | 166 |
| M20 | 625 | 165 | 135 | 0 | 640 | 196 | 500 | 160 | 22.2 | 166 |
| M30 | 625 | 165 | 135 | 0 | 560 | 294 | 500 | 160 | 22.2 | 166 |
| E3M20 | 606 | 160 | 132 | 28 | 640 | 196 | 500 | 160 | 22.2 | 166 |

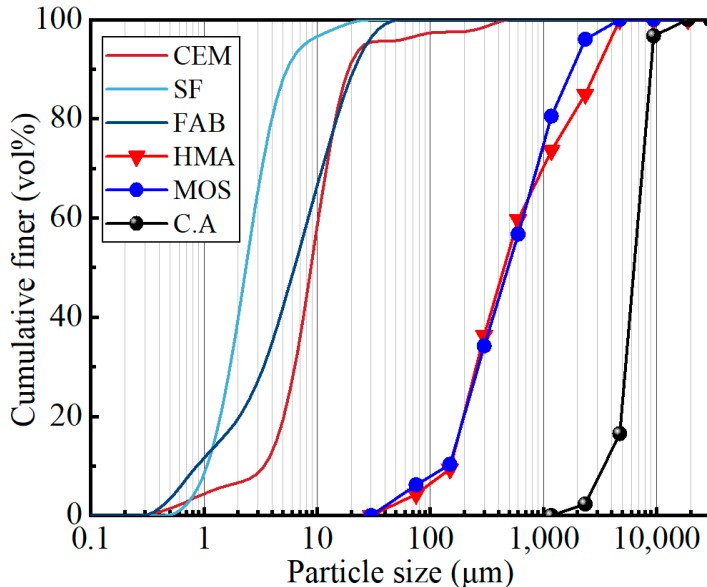

**Figure 1.** Particle size distribution of the cementitious materials and aggregates.

*2.3. Experiment Methods*

Preparation Process: MOS was introduced into a 200-mesh container and submerged in a water tank for 24 h to achieve a saturated surface dry state, after which it was stored. During the mixing process, the components of the raw materials were weighed in accordance with the prescribed ratios. Firstly, we dry-mixed the cementing material and fine aggregate for 1 min, added water-reducing agent and 75% water, continued to mix for 3 min to the flow state, evenly scattered the steel fiber, added the remaining water to continue mixing for 2 min, and finally added the coarse aggregate and stirred until uniform. The total mixing time was controlled at about 12 min. Afterward, the molding was placed on the shaking table for 30 s and covered with cling film for one day. After demoulding, normal maintenance procedures were followed, and relevant performance tests were conducted at the 28-day mark. Test method: The workability of UHPC was measured according to GB/T50081-2019 [25]. Compressive strength tests, flexural strength tests, and elastic modulus tests of UHPC were carried out in accordance with GB/T31387-2015 [26]. In addition, the early autogenous shrinkage deformation of UHPC was tested based on GB/T50082-2009 [27]. The early hydration process of UHPC was studied using an eight-channel hydration isothermal calorimetry (TAM Air, TA Instruments, Thermo, Waltham, MA, USA). The morphology of the aggregate microareas was observed by a QUANTA FEG 450 scanning electron microscope (Zeiss, Oberkochen, Germany). The TI-980 nanoindenter instrument was used to test the micromechanics, and the loading program was selected as trapezoidal loading with a loading rate of 200 μm/min increasing linearly to 2000 μm. Then, the load was held for 10 s to reduce the error due to creep and then reduced at the same rate. In addition, the spacing of indentation points was selected to be 5 mm in order to avoid the interference between the two indentation points. The experimental procedure is shown in Figure 2 below.

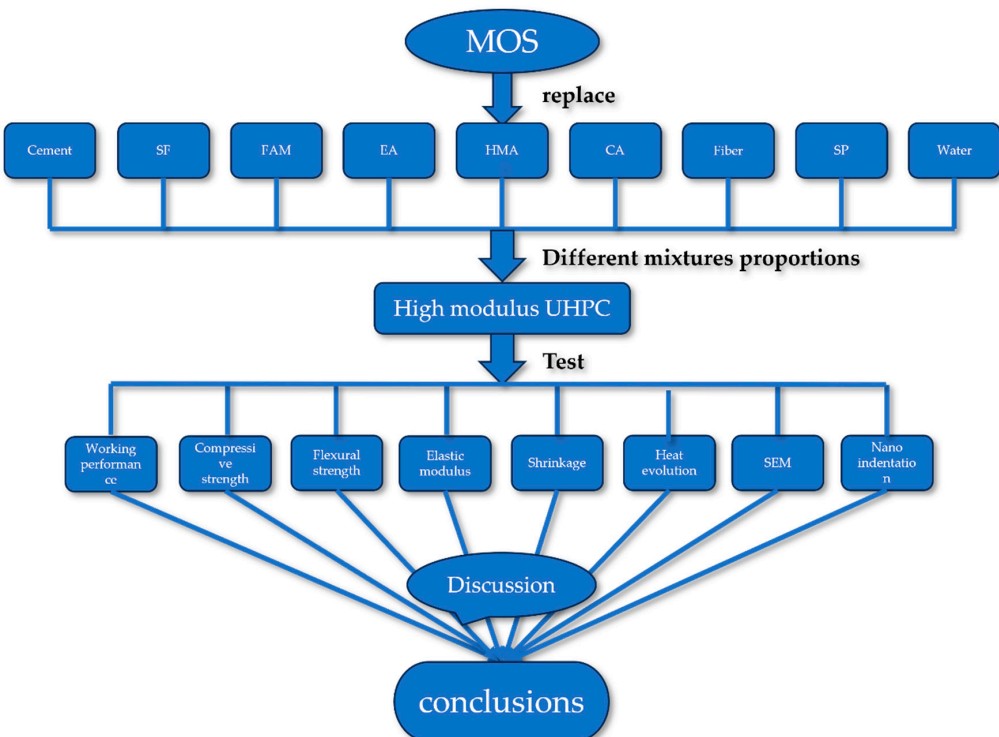

**Figure 2.** Experimental flowchart of the coupled effect of expansion agent and internal aggregate on shrinkage of high-modulus UHPC.

## 3. Results and Discussion

### 3.1. Working Performance

The workability of the high-modulus UHPC is shown in Figure 3. It can be found that the addition of EA had no effect on the flow of UHPC mixes, whereas the incorporation of prewetted MOS can effectively improve the flow of the mixtures. Meanwhile, the flow of the M30 group increased by 34.9% compared to the plain UHPC (N group). This is because of the porous surface of MOS and its excellent water storage performance. During the mixing process, the saturated MOS releases partial water during the mixing process, which increase the free-flowing water in the mix and improves the working performance of UHPC. However, the higher actual water–cement ratio affects the mechanical properties of UHPC when the MOS doping is too large. Therefore, taking into account the mechanical properties and working performance, the E3M20 group is considered to be the optimal ratio (scales 27.9% better than the N group).

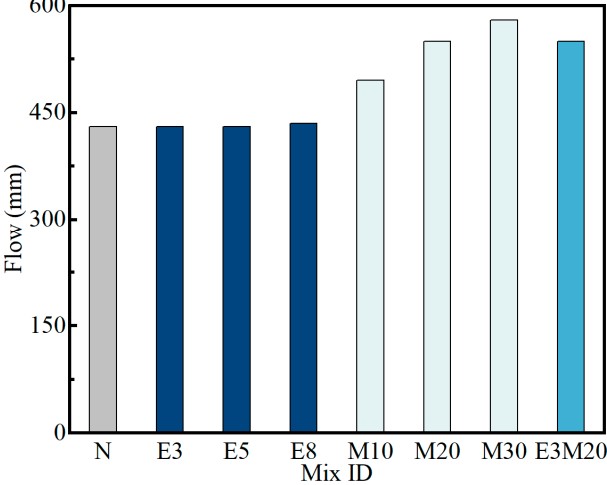

**Figure 3.** Flowabilities of UHPC.

*3.2. Compressive Strength*

The compressive strengths of UHPC at different stages are shown in Figure 4. It can be found that the addition of EA and ICA has a significant impact on the compressive strength of UHPC. With the increase in proportion of CaO–MgO composite EA, the 28 d compressive strength of concrete decreases, but its early compressive strength increases. Compared with plain UHPC, the 3 d (The term 'concrete at 3 days' refers to concrete that has undergone 3 days of curing and solidification.) compressive strength of group E8 increased by 10.1 MPa, but the 28 d compressive strength decreased by 11.3 MPa. This is due to the fact that the change in compressive strength is related to the EA affecting the hydration process of the cementitious slurry. Previous research has similarly shown that the expansion stresses generated by the EA reaction also increase the porosity of the matrix, leading to a looser internal structure and a reduction in the 28 d compressive strength [28]. In addition, the 3 d compressive strength of UHPC decreases by 3.8 MPa, 8 MPa, and 9.5 MPa as the doping amount of MOS increases from 0 to 30%. With the moisture difference formed by the hydration of the slurry, the internal water of MOS is slowly released to improve the internal relative humidity of slurry. Simultaneously, the secondary hydration of the cementitious slurry occurs to compensate for the reduction in strength due to the increase in the actual water–cement ratio. In addition, compared with the 3 d strength, the 28 d compressive strength of the N, M-10, M-20 and M-30 groups increased by 86.6%, 88.9%, 91.3%, and 77.7%, respectively. A significant decrease in the compressive strength of the M30 group may be attributed to the fact that the higher MOS admixture increased the actual water–cement ratio of UHPC dramatically, whereas its internal curing effect could not compensate for the decrease in concrete strength.

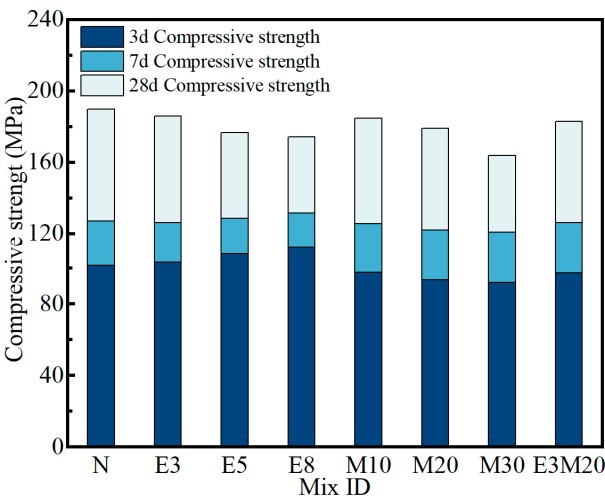

**Figure 4.** Compressive strength of UHPC at different stages.

*3.3. Flexural Strength*

The flexural strength of UHPC is shown in Figure 5. The incorporation of EA and ICA could improve the flexural strength of UHPC. Specifically, the flexural strength of group E3 was increased by 1.4 MPa compared with group N. But the flexural strength of UHPC decreased to 19 MPa when the dosage of EA continued to increase to 8%, which might be related to the interfacial bond strength of the substrate. Le [23] has shown that the reduction in UHPC shrinkage could enhance the interfacial bond strength of the matrix. Therefore, when the amount of EA was 3%, the decline in the compressive properties of UHPC was low and the flexural strength was increased. However, too much EA affected the C–S–H gel content resulting from the hydration of the slurry, which could damage the flexural properties of UHPC. In addition, the inclusion of MOS improved the flexural strength of UHPC, but the flexural strength of the M30 group decreased by 1.4 MPa compared with the M20 group. This is mainly due to two reasons: (1) the addition of the saturated prewetted

MOS reduces the viscosity of the slurry, and the steel fibers can be more easily adjusted in their distribution and orientation during mixing and forming [29]; (2) the internal curing effect of MOS can effectively improve the interfacial bonding properties of the substrate. However, the matrix property degradation caused by the high actual water–cement ratio at 30% MOS doping can also weaken the flexural strength of UHPC.

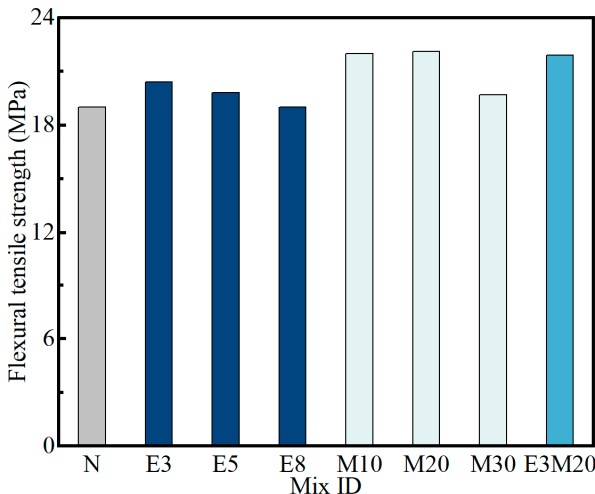

**Figure 5.** Flexural strength of UHPC.

*3.4. Elastic Modulus*

The elastic modulus of high-modulus UHPC is shown in Figure 6. It can be clearly seen that the elastic modulus of the prepared UHPC was higher than 50 GPa, which is mainly due to the use of high-elastic modulus aggregate basalt and HMA. The elastic modulus of UHPC would be reduced by either the inclusion of an EA alone or the ICA. On the one hand, the expansion reaction of the EA affected the hydration degree of the slurry, which also made the internal structure of UHPC looser. Therefore, as the amount of expansion agent dosing increased (while dosing alone), the more noticeable the decrease was in the modulus of elasticity of UHPC. On the other hand, the internal curing effect of MOS was hardly enough to counteract the decreasing effect of the increase in the actual water–cement ratio of UHPC and the decrease in the proportion of HMA dosing. As a result, the modulus of elasticity of the M30 group was significantly reduced. In addition, the insignificant decrease in the elastic modulus of the M10 and M20 groups was due to the fact that the porous and high-strength MOS could effectively improve the interfacial bonding properties of the matrix by means of internal curing.

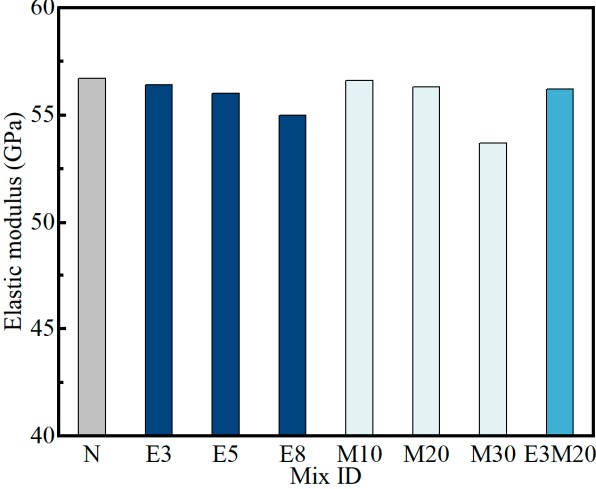

**Figure 6.** Elastic modulus of UHPC.

### 3.5. Shrinkage Behavior

The shrinkage of each group of UHPC is illustrated in Figure 7. It can be seen that the 180 d shrinkage of UHPC gradually decreased with the increase in the dosage of expander (i.e., 211 µε, 281 µε, and 309 µε compared to plain UHPC). Therefore, the EA compensates for UHPC shrinkage with significant effect, but because of the lack of moisture inside the UHPC, the high dosage of EA could not be fully hydrated and its role in compensating for the contraction is limited. In addition, the 180 d shrinkage of UHPC decreased from 596 µε to 373 µε when the prewetted MOS doping was elevated from 0 to 30%. The reason for this compensated shrinkage is that the prewetted MOS with small and interconnected pores on its surface can save a certain amount of water, which will not be released in the mixing. Then, with the hydration and self-drying of the cement paste, at this time, the humidity difference between the prewetted MOS and the surrounding paste, a large amount of water in the capillary pores is consumed and the pressure in the capillary pores is reduced, forming negative capillary pressure. Finally, the absorption force generated by the negative pressure gradually releases water from the pores of the prewetted MOS, promotes the hydration of the cementitious material around the aggregate, and reduces the shrinkage stress of the capillary pores, thus compensating for the shrinkage deformation of the concrete at all stages.

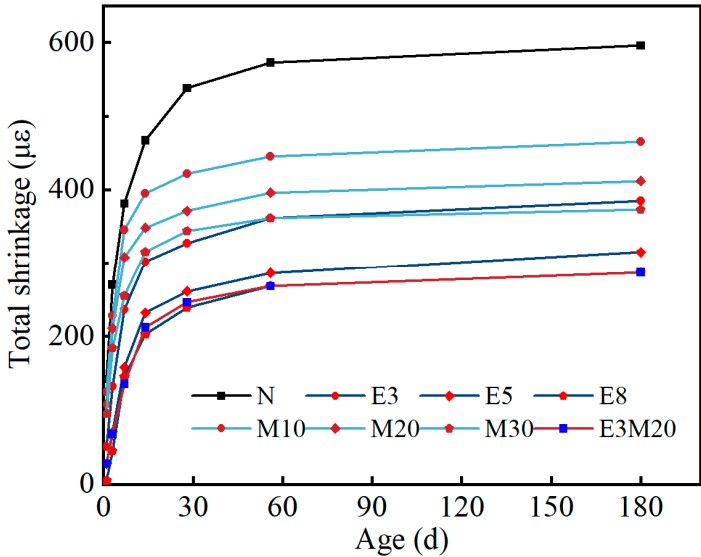

**Figure 7.** The evolution of UHPC shrinkage with time for each mix proportion.

### 3.6. Heat Evolution

The effect of different amounts of EA on the hydration process of UHPC is shown in Figure 8. From the exothermic rate curve, it can be seen that the UHPC slurry went through an induction period of 6 h. After the dosing of EA was gradually increased, the heat flow during the induction period of slurry hydration exhibited a gradual increase. After that, the induction period is followed by a period of accelerated hydration for about 12 h. In this period, the exothermic rate of slurry hydration reached its peak, which was followed by a deceleration period lasting about 28 h. Finally, the hydration of the cement paste was essentially stable and the exothermic rate steadily decreased [30]. From the figure, it can be learned that with the increase in the dosage of the EA, the peak exothermic rate of cement slurry hydration, the peak appearance time, and the 72 h exothermic rate all show a trend of increasing and then decreasing. When the admixture of the EA was raised from 0 to 5%, the hydration of the CaO group in the EA accelerated the hydration exothermic rate and peak time of the cement paste and the accumulated exothermic heat gradually increased. However, when the amount of EA reached 8%, the hydration of the CaO group made the free water inside the slurry decrease rapidly, which slowed down the exothermic rate and

peak time of slurry hydration, and the slurry hydration showed a longer induction period and acceleration period. This indicated that the excessive admixture of EA resulted in a significant delay of slurry hydration and a fallback in cumulative heat release.

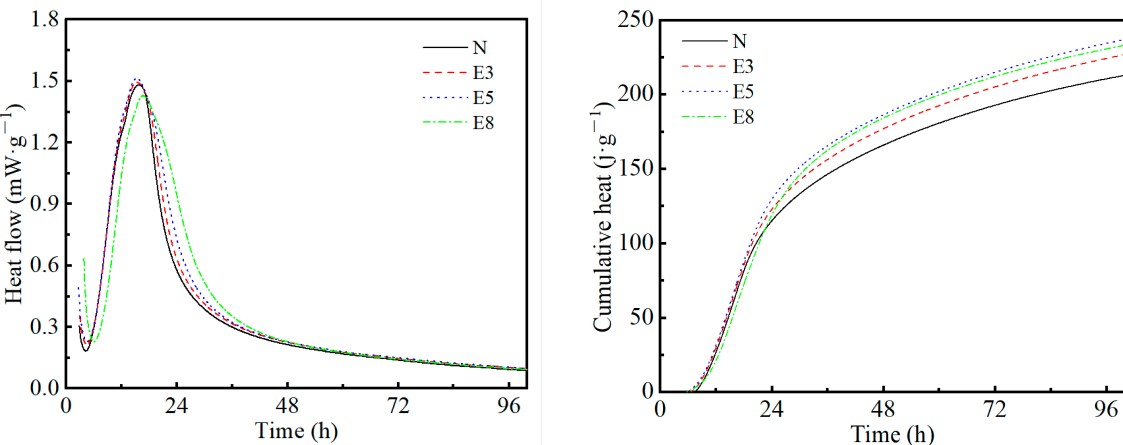

**Figure 8.** Effect of EA on the heat evolution of UHPC.

### 3.7. SEM

The surface morphology of the UHPC combined with the aggregate is exhibited in Figure 9. As Figure 9a demonstrates, the interfacial transition zone between MOS and cement paste, it can be seen that the surface of MOS was rough and porous, and the cement paste is tightly connected to the aggregate; thus, the interfacial bonding performance was excellent. Moreover, the MOS aggregate surface is uneven, and the unhydrated cementitious particles and hydrated paste would gradually fill the pores and grooves on the aggregate surface during the mixing process of UHPC so as to make up for the increase in matrix porosity brought about by the porous aggregate. Furthermore, as the concrete hardens, the hydration products grow inside the aggregate, and it can become the lock that connects the aggregate to the matrix, making the aggregate and the slurry tightly anchored into a single unit. In addition, the water released from the pores of the aggregate would promote hydration of the interfacial transition zone and improve the homogeneity of the matrix. In addition, the transition zone of the interface between HMA and matrix is shown in Figure 9b. Due to the microscopic pores on the surface of HMA aggregate, it would absorb the free water within the mixture during the mixing process. Cement slurry and some unhydrated cementitious particles were tightly attached to the surface of HMA, resembling a matrix, thus forming a dense interfacial transition zone.

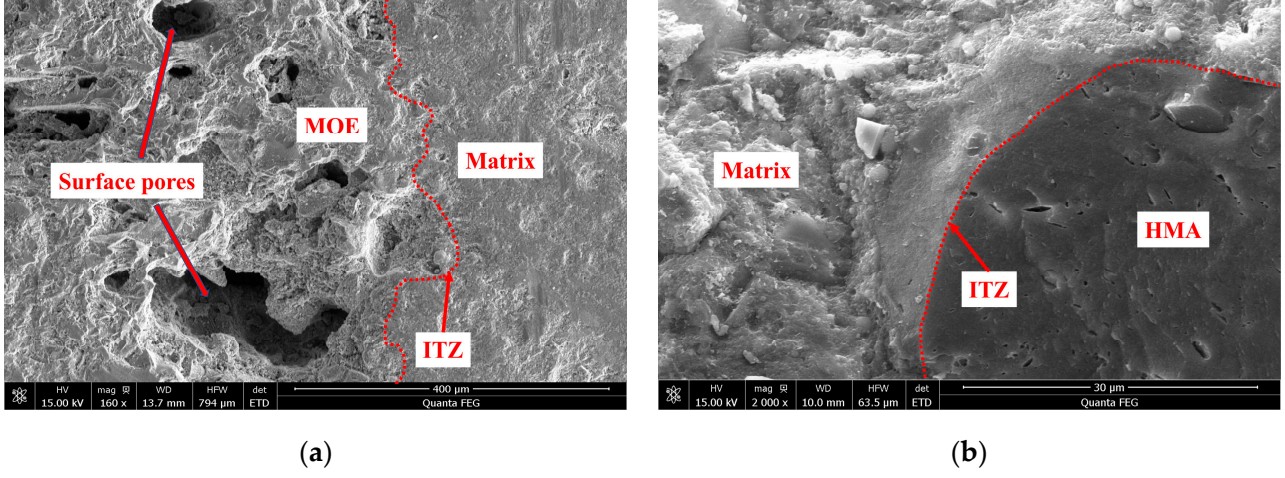

**Figure 9.** SEM images of the UHPC sample. (**a**) MOS and (**b**) HMA.

*3.8. Nanoindentation*

Typically, the elastic modulus of the main components of the slurry is porous, with the elastic modulus below 10 GPa; low-density (LD) calcium silicate hydrate (C–S–H) gel, with elastic modulus of 10–20 GPa; high-density (HD) C–S–H gel, with elastic modulus of 20–30 GPa; ultrahigh-density (UHD) C–S–H gel, with elastic modulus of 30–50 GPa; and unhydrated gelled particles, with elastic modulus greater than 50 GPa [31]. The results of the nanoindentation test are illustrated in Figure 10, and Figure 10a shows the elastic modulus of each indentation point (each group of indentation points contains aggregate, interfacial transition zone, and matrix). In summary, the calculated hardness obtained for each indentation point is shown in Figure 10b.

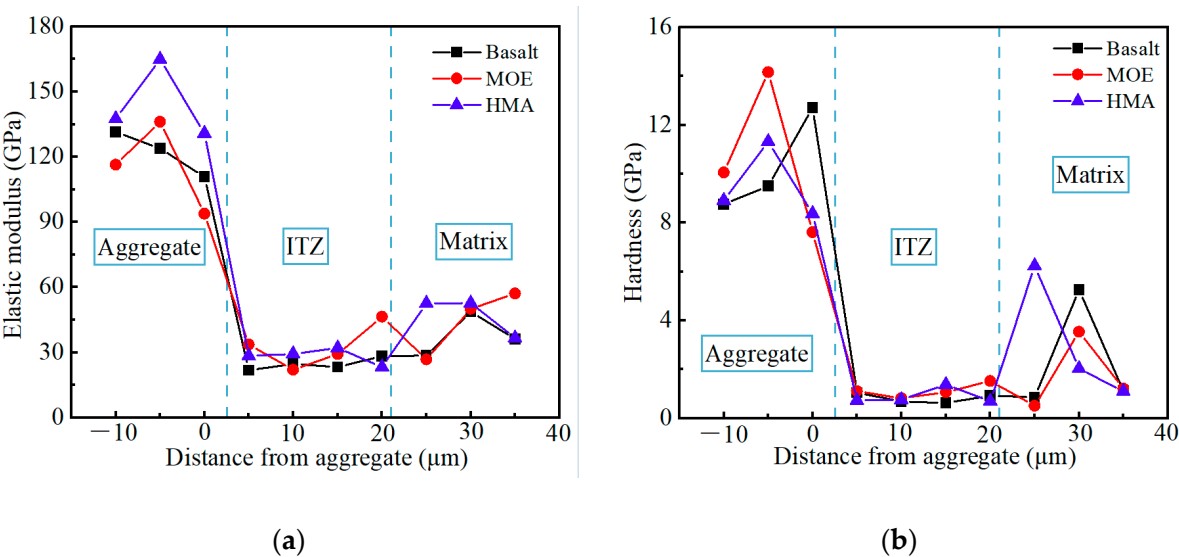

**(a)**																													**(b)**

**Figure 10.** Nanomechanical properties of indentation points. (**a**) elastic modulus and (**b**) calculated hardness.

In Figure 10a, it can be found that even though MOS has a porous structure, the aggregate itself still has excellent mechanical properties, and the modulus of elasticity of the three types of aggregates: basalt, MOS, and HMA are 123.85 GPa, 136.08 GPa, and 164.82 GPa, respectively, of which the nanoelasticity modulus of MOS was slightly reduced compared with that of HMA. Regarding the interfacial transition zone (ITZ), the average modulus of elasticity in the ITZ for the three aggregates of basalt, MOS, and HMA are 24.58 GPa, 28.32 GPa, and 28.29 GPa, respectively. In contrast to fine aggregates, the coarse aggregate ITZ had a lower modulus of elasticity. This occurs because concrete tends to accumulate water beneath the coarse aggregate prior to setting, creating a water-filled region. At the same time, the actual hydrogel around the coarse aggregate was relatively large, weakening the strength of the coarse aggregate ITZ [32]. However, the nanomechanical properties of the ITZ with the matrix under the effect of MOS internal maintenance were also not significantly decreased (the enhancement of the actual water–cement ratio) compared with HMA. Therefore, when the replacement dose of prewetted MOS was 20%, the better aggregate strength and interfacial bonding properties of MOS compensated for the decline in mechanical properties caused by the increase in the actual water-to-cement ratio of UHPC and the decrease in the dosage of HMA. The average elastic modulus of the cement paste is 43.29 GPa, which indicates that the cement paste of UHPC is mainly composed of HD C–S–H gels and UHD C–S–H gels. In addition, some indentation points in the cement paste have a high elastic modulus, which may be unhydrated cementitious particles in the lower water-to-cement ratio environment of UHPC. Therefore, the incorporation of porous high-strength aggregate MOS does not lead to a significant reduction in the strength of UHPC.

### 4. Conclusions

(1)　The expansion reaction of the expansion agent (EA) can effectively compensate for the shrinkage of UHPC, but the expansion agent has high hydration activity in the early stage, which leads to the lack of internal moisture; i.e., the expansion effect can not be fully exerted, and the slurry is not sufficiently hydrated. In addition, the compressive strength and elastic modulus of UHPC slightly decreased, and the flexural strength first increased and then decreased.

(2)　The internal curing aggregate MOS can effectively mitigate the shrinkage of UHPC. Moreover, the porous surface of the aggregate makes it possible to release water slowly inside the material after prewetting and improve the interfacial properties of the aggregate and the matrix by internal curing. As a result, its better aggregate strength and interfacial bonding properties compensate for the decline in mechanical properties caused by the increase in the actual water-to-cement ratio of UHPC and the decrease in the HMA dosage.

(3)　The volume stability of UHPC can be maximized through the synergistic effect of expansion and contraction of the EA and the compensatory contraction of the ICA Moreover, the decrease in compressive strength and modulus of elasticity of UHPC is not significant, and it is also effective in improving its flexural strength.

**Author Contributions:** Methodology, Data curation, and Writing—original draft, M.Z.; Methodology, Funding Supervision, and Writing—review and editing, J.L.; Methodology and Writing—review and editing, D.C.; Methodology and Data curation, T.Y.; Methodology and Data curation, Z.Y.; Methodology and Data curation, B.Q.; Methodology and Data curation, W.Q.; Methodology and Data curation, B.L.; Methodology and Data curation, B.S.; Methodology, Data curation, and Software, C.Z.; Project administration and Resources, L.G.; Data curation, Y.L. All authors have read and agreed to the published version of the manuscript.

**Funding:** This work was supported by the Science and Technology Project of Foshan Transportation Science and Technology Co., Ltd. (FJK (2020) B-062).

**Institutional Review Board Statement:** Not applicable.

**Informed Consent Statement:** Not applicable.

**Data Availability Statement:** The data presented in this study are available on request from the corresponding author.

**Conflicts of Interest:** The authors declare no conflict of interest.

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
