# Peer review of "Coupling Effect of Expansion Agent and Internal Curing Aggregate on Shrinkage of High-Modulus Ultra-High-Performance Concrete"

_coatings, doi:10.3390/coatings13091571_

Round 1

Reviewer 1 Report

The paper presents a study on the Ccoupling effect of expansion agent and internal maintenance

aggregate on shrinkage of high modulus UHP.

The following recommendations are proposed:

·      Please split the introduction into two sections: introduction and literature review.

·      Please, provide a flow chart of the paper organization.

·      Overall, English needs to be double-checked for typos.

·      The conclusions section can be better organized by including a short summary.

Main Concern:

What is the innovation that this paper brings into scientific knowledge?

Spelling check

Author Response

Dear reviewer

      I am thankful for the suggestions you have provided for the article. I have made the necessary revisions based on your advice, and the modified content has been attached herewith. Once again, I express my gratitude for the time and effort you have dedicated to this article. With sincere regards, I wish you success in your endeavors and good health.

Reviewer 2 Report

The article is of interest to scientists engaged in the materials science of concrete. The article contains interesting results on its drying and mechanical properties. However, the English of the article requires serious revision, especially in the last chapters. There are a lot of unfinished sentences and grammatical errors. In addition, many abbreviations are used that are not deciphered. A list of all comments is given file attached.

English is poor. Many sentences are not finished. Sometimes it is difficult to understand.

Author Response

(The authors gave the same response as above.)

Reviewer 3 Report

The paper is not written according to the requirements. The authors are recommended to systematize the results in such a way as to highlight the objectives of the research carried out and in a clear way the obtained performances.

There are many mistakes in writing in English, sentences that do not say something, an some are given below in lines:

38: To maximize the excellent mechanical properties of UHPC [2].

39:  Self-shrinkage is defined as a macroscopic volume change caused by capillary pressure due to the chemical process of self-drying and cement hydration, and there is no water transfer to the surroundings [4]. 

68: The expansion of CaO-based and CaO-based occurred early with rapid development and could stabilize by 28 days.

101: Valipour [24] claimed that the shrinkage of UHPC prepared with 7.5% expansion agent and 60% pre-wetted light aggregate is only 273 με at 28 d.

120: The fly ash microbeads 28d activity index is 113% and the water demand ratio is 102%.

129: Flat microfine copper-plated steel fibers with 0.2mm equivalent diameter, 13mm nominal length and 130 7850 kg/m3 density.

The article must be rewritten.

Extensive editing of English language is required.

Author Response

(The authors gave the same response as above.)

Round 2

Reviewer 2 Report

The paper is revised. Many questions are answered and corrected. However, English need in moderate improvement. The paper can be accepted after English editing. If it will be send for reviewing again, please, send in Word format. The text in pdf-format with track-corrections is very difficult to read.

English still needs editing. The text contains a number of typos (for example, "shouwn") and unclear sentences. For example, lines 4223-428: it is written that the elastic modulus is pore, HD gel, UHD gel etc. What it means?

Author Response

Dear reviewer

      I am thankful for the suggestions you have provided for the article again. I have made the some revisions based on your advice, and the modified content has been attached. Once again, I express my gratitude for the time and effort you have dedicated to this article. With sincere regards, I wish you success in your endeavors and good health.

Reviewer 3 Report

Some writing errors need to be corrected: the repetition of some words in the same sentence. It is also necessary to respect a unique form for use in the text: Figure or Fig.?

Some writing errors need to be corrected: the repetition of some words in the same sentence. 

Author Response

(The authors gave the same response as above.)
